# Modelling Urban Tourism in Historic Southeast Asian Cities

**Jackie Lei Tin Ong [1],\* and Russell Arthur Smith [2],\***

[1]    Tourism and Hospitality Management, RMIT University, Hanoi, Vietnam
[2]    Sitetectonix Pte Ltd., Singapore 189969, Singapore
\*    Correspondence: leitin.ong@rmit.edu.vn (J.L.T.O.); russellarthursmith@stxla.com (R.A.S.)

**Abstract:** Older cities with significant physical historic resources have become increasingly important centers for urban tourism, though contemporary attractions and events are often important in their own right. It is acknowledged that urban areas are multifaceted agglomerations where tourism complements other economic sectors and competes with them for limited resources. A limited investigation of the temporal dynamics of change of tourism in cities has been undertaken in the several countries in Europe and North America. Much less has been studied for the countries of Southeast Asia where tourism has expanded rapidly over the past several decades, a trend that is projected to continue. Urban tourism has and will continue to be important in Southeast Asia. This paper reports on the spatial modeling of the evolution of tourism in two historic cities in Southeast Asia that seeks to better understand the dynamics of temporal change of tourism within their respective urban contexts. The cases of Bangkok and Jakarta have been compared with the Ashworth and Tunbridge model to identify conformity and otherwise of Asian cases with the European theory.

**Keywords:** modeling urban tourism; Southeast Asia; historic cities; Jakarta; Bangkok; evolution of tourism

## 1. Introduction

Tourism in the Asia and the Pacific region has expanded by 7.1% annually from 2009 to 2019, which is the highest rate of growth globally by region. This compares with the annual average of 5.1% [1]. In the Southeast Asia region, travel and tourism contributed 12.6% of the regional economy and provided 12.2% of the employment in 2018 [2]. Urban tourism in Southeast Asia is a significant component of this and many tourists are lured to the historical cities due to their rich culture, heritage, and multifunctional nature. Facilities are not solely produced or consumed by tourists but are shared by both residents and tourists [3,4].

As noted by the United Nations [5], the world is urbanizing. In total, 54% of the world's population lived in urban areas in 2015, a number that is expected to increase to 60% by 2030. Postma, Buda, and Gugerell [6] (p. 95) highlighted the importance of tourism in urban areas by noting that " ... city tourism has consistently been one of the fastest growing segments of the travel phenomenon in countries with developed economies." This observation takes on greater stature when the modernization of many Global South countries has occurred during this period with the attainment by some of developed-country status, especially in Asia [7]. The modernization of cities comes with many urban problems related to slums that lack sanitation and water supply, poor public services, and inadequate public transportation resulting in traffic congestion, to mention a few [8]. For many Asian cities, tourism has a significant economic component. The Euromonitor International surveyed 600 global cities and ranked the top 100 destination cities by international tourist arrivals in each city, regardless of point of entry to the respective countries [9]. For these ranked cities, 41% are in Asia and 32% in Europe, with the remainder in Africa, the Americas, and the Middle East, which underlines the importance of international tourism for Asian cities.

Modelling the spatio-temporal dynamics of tourism in urban areas is important for a deeper construct of the interactions between urbanization and one of its economic drivers: tourism. Such modelling may assist urban and tourism planners to gain an understanding of how tourism develops with the growth of cities. This modelling would not be predictive but could provide some insights to assist planners with their forward decision-making. Although urban tourism has become a focus for some research, there has been limited investigation of the temporal dynamics of change of tourism in cities in the Southeast Asia context where tourism has expanded rapidly over the past several decades. This paper thus reports on research related to two historic cities of Southeast Asia, Jakarta, and Bangkok.

The Case of Bangkok. Originating in the 15th century as a small trading port of the Ayutthaya kingdom that was seated to its north, Bangkok has become a major city in Thailand and Southeast Asia. The city is now the center for national government and the seat of the Chakri dynasty. Following the Pacific war of the middle of last century, the city grew rapidly and now has a greater Bangkok urban region population of more than 14 million. The greater part of Bangkok is governed by the Bangkok Metropolitan Administration, which has 50 Districts. Because of its size and national importance, there are 15 administrative departments under the city Governor [10–12].

The Case of Jakarta. Originating as Sunda Kelapa during the Sudan kingdom, Jakarta has been continuously inhabited since the fourth century. From an estuarian trading port, Jakarta has become the capital of Indonesia and its largest city with a greater urban population of more than 35 million. During the Dutch colonial period, the city was the effective capital for the colony, known as Batavia. Presently, the national government is in Jakarta. The city is a province and is administered through five cities and one regency. A recent national policy is to relocate the national government to the province of East Kalimantan on the island of Borneo [11–13].

Commencing with feudal monarchies, many Asian cities have developed unique urban characteristics. Both of the cases that we have studied, which are in different Southeast Asian countries, have experienced significant and intense urban growth in recent decades. This growth has generally been driven by a multiplicity of economic sectors. The post Pacific-war period, which occurred in the second-half of the 20th century, provided opportunities for these countries to propel themselves forward into a new modernity where tourism has been a significant economic activity. In 2018, the travel and tourism sector contributed 1,213,000 jobs to Jakarta, which was 8.5% of the total jobs in this city and 3.1% of its total GDP. For Bangkok, the contributions were 682,000, 7.2%, and 10.6%, respectively [2].

The urban spatial and temporal patterns for the development of Jakarta and Bangkok are compared with the Ashworth and Tunbridge [14] model. The general theory on urban tourism has an extensive representation in the literature [3,15]. As Ashworth and Page [16] (p. 13) note, "Cities have become increasingly engaged in profiling themselves within global competitive arenas" where tourism has become a major driver for attracting increased tourism demand. While tourists visit cities for a wide range of reasons, the attraction of historic resources is an advantage that cities frequently maximize for their product development and marketing advantage [17]. Urban tourism here refers to tourism that occurs in major urban centers, that is cities. Large cities often have multiple functions that include several significant economic sectors. These may include finance, manufacturing, healthcare, transportation, education, real estate, and tourism, among others [3,18]. These cities provide opportunities for economic development that benefit from interdependence of these sectors and frequently leverages of existing infrastructure. Primate cities [19] are typically gateways for their countries, which is important for the development of their tourism sectors. Airlines and cruise lines deliver tourist to these gateway cities where they have the potential for economic impact.

Tourism has the potential for both positive and negative economic, social, and environmental impacts [20]. Such impacts have been well documented where the negative impacts in cites and at historic sites include congestion, gentrification, and resident dis-

content [18,21,22]. One major driver has been mass tourism, which has been deemed to be overtourism in some urban destinations [23]. It is also acknowledged, in the context of the present research, that tourism in developing countries—least and less developed countries—has unique aspects not always found in developed economies [24]. The management of tourism is often difficult but in cities it is especially so because of the complex interrelations with other large-scale functions and competition with other economic sectors for resource and political power [25]. Nevertheless, Getz [26] notes that tourism development within urban areas often has distinctive spatial form and temporal dimension.

Here, we seek to compare two cases, specifically the mega-cities of Bangkok and Jakarta, with a well-known model of the evolution of tourism in historic cities. We do this to gain insights of the variations between the generalized evolution of historic European cities as found in the Ashworth and Tunbridge model and those which are found in the historic cities of Southeast Asia. Utilizing the four-phase structure of the Ashworth and Tunbridge model as an analytical framework, comparisons between the model and the cases are presented, below, phase by phase. Analytical graphic representations are also presented.

## 2. Modeling Urban Tourism

One urban tourism model was developed by Smith [27]. The model describes the transition of beach resorts from natural to urban contexts. This model was developed from the study of four case cities in the Asia Pacific region. Parallel work for islands is found in Weaver's [28] model of urban tourism for small Caribbean islands. Smith's model is useful for large-scale seaside developments where tourism is the dominant economic sector. It is of limited use for urban contexts where there may be a range of dominant economic sectors, of which tourism is only one.

Tourism in historic cities has become increasingly important in recent years [29] though the attraction of historic cultural resources in cities is not new; spanning millennia [30]. Historic urban features have also been applied for the generation of tourism demand and the revitalization of historic cities [31]. Unlike cities dominated by their tourism sector, such as beach resorts, many historic cities have more complex economies. Tourism in these forms of urbanization has been modeled by Ashworth and Tunbridge [14] but for medium-sized and long-established historic European cities. This model has four phases of tourist-historic city development. Initially, for Ashworth and Tunbridge, there is commercial development in the center of the original city and a small central business district (CBD) is created, which is entirely within the historic city. This is centered on the oldest district of the city. Over time, there is some conservation of the historic city with partial migration of the commercial city. CBD activity moves outside of the original CBD where newer functions of retail, commerce, and administration develop, which is the second stage of this model. Thus, an historic city forms, which is the central and oldest part of the city and includes part of the CBD. At the third stage, a tourism city develops overlapping part of the historic city and commercial city. Subsequently, at the fourth stage, there is an expansion of historic and commercial cities in much of the original city with progressive gentrification of housing. Burton [32] extended the Ashworth and Tunbridge model by adding a leading stage for a five-stage model, which defines the original city without a CBD. As has been noted by Shin [33], tourism at these historic cities is influenced by the historic culture of these destinations and in turn influences spatial pattern of the urban tourism there.

The Ashworth and Tunbridge model was chosen for our research as it incorporates key indicators that allow for the integrated analyses of tourism development within urban contexts over long periods. While a range of data are considered, the spatial dimensions of change over a time provide useful insights as to their dynamics. As city planning is grounded in the physicality of spatial constraints, the graphic representations provide both analytical and communicative avenues of value.

The Ashworth and Tunbridge research bias is clearly European-centric that does not address urban tourism's spatial patterns as found in other regions of the world. Subse-

quently, Ashworth and Page [16] surveyed the literature and identified a range of foci for urban tourism research, arguing that greater attention needed to be placed on social science research in this field. While this is an admirable observation, a central factor for urban tourism is the spatial dimension as it determines, along with other factors, where the actual tourism economy functions. This informs municipal policy makers and physical planners for their plans for urban development and management.

Some researchers have extended the original model by Ashworth and Tunbridge through consideration of a number of cases in non-European regions. Oppermann and Din [34] published an extension of this work for urban hotel development and evolution in Kuala Lumpur, Malaysia, which is a medium-sized city. Five stages of development were identified. Li et al. [35] identified the spatial pattern of hotels in Hong Kong, a large city in China, and described how this pattern changed over time. Commercial land types and the number of nearby attractions influenced the spatial pattern of upper-grade hotels in Hong Kong. De Bres [36] discussed the applicability of Ashworth and Tunbridge's classification in the context of then recently established communities in Kansas and found that it differed significantly from the European theory. This was, perhaps, a result of the small size of these communities, which had not then developed significant clusters of tourist services such as accommodation.

## 3. Method and Objective

This research seeks to describe the general spatial and temporal development of tourism as found in two major cities of Southeast Asia. It focuses on the spatial dynamics of the tourism facets, as defined by Ashworth and Tunbridge, within overall urban growth for these case cities. The research relies extensively on case evaluation method [37,38] and comparative evaluation method [39]. Examples of the application of these research methods in tourism include Simpson and Wall [40] and several studies cited in the references below. Here, we summarized aspects of the case evaluations for Bangkok, Thailand and Jakarta, Indonesia. These cases were compared with the established Ashworth and Tunbridge spatial temporal model of urban tourism, as modified by Burton. The objective was to identify similarities and differences of the two Asian cases and how they compare with the European-based model.

Bangkok and Jakarta are the primate cities for their respective countries and have been for some decades [41]. Both are also the capitals of Thailand and Indonesia, respectively, and have populations in excess of 10 million [11]. Jakarta has a recorded history spanning more than a millennium. Its more recent history includes Dutch and Japanese colonial periods before the independence of Indonesia was fully gained in 1949. During the early part of Dutch rule, there was a walled city, Batavia, related to the port on the Ci Liwung river. Established under the rule of the Ayutthaya kingdom, Bangkok was a river customs station more than six centuries ago. Bangkok has never experienced colonial administration. Interestingly, both Jakarta and Bangkok have distinct, largely unrelated, monarchal histories.

Historic sites have long been destinations for tourists, where these places have additional attraction when they possess buildings related to their historical significance [42,43]. Tourists are attracted to places that are historically important and especially so if they can view and enter buildings related to that history. Should these sites be within urban areas, their potential for tourism is often enhanced because of proximity to existing transportation, accommodation and other supporting infrastructure [44]. Governments actively protect and conserve their urban historic districts for their cultural heritage value as well as tourism-sector economic and employment benefits where there is a symbiotic or mutually supportive relationship between conservation and tourism [20].

Tourism development in the historic districts of cities also has the potential for negative impacts such as overtourism, gentrification, and host-visitor conflicts [45]. Distortion of the historic value of the place that devalues the indigenous culture through homogenization can also occur [46]. The tensions of change in historic cities have been highlighted by

Palacios, Mellado, and León [47] in Barcelona, Spain where traditional neighborhoods such as Marina del Prat Vernell are vulnerable to urban renewal for commercial and tourism functions. Anticipation of and defending against these dis-benefits demand proactive municipal planning and management to which, it is hoped, this research will contribute in a small way.

We hope that the contribution of our research to the general knowledge about tourism, cities, development and space in Global South cities is a modest extension of the understanding of existing theory as proposed by Ashworth and Tunbridge (and extended by others) in the Southeast Asian region, especially as the two case cities include periods of rapid urban growth.

Findings from our research may be useful for urban planning in Bangkok and Jakarta. Local governments, working with other governmental agencies and relevant private sector organizations, could apply historic understandings as well as trends towards strategic planning policy formulation. Decision making for implementation of physical plans may also be facilitated and justified.

## 4. Findings

Comparative spatial analyses of cases with a theoretical model, at the scale of a city, present unique challenges. Here, the Ashworth and Tunbridge model is a considered generalization, sans distinctive features of its formative specific cases. Yet the cases of Bangkok and Jakarta, both megacities with populations exceeding 10 million, have evolved over many centuries where their natural geographical features have exerted major influences on their respective evolutions and particularly their physical urban forms.

The findings are presented in graphic format in Figures 1–5, with pertinent points highlighted below.

*Stage One*

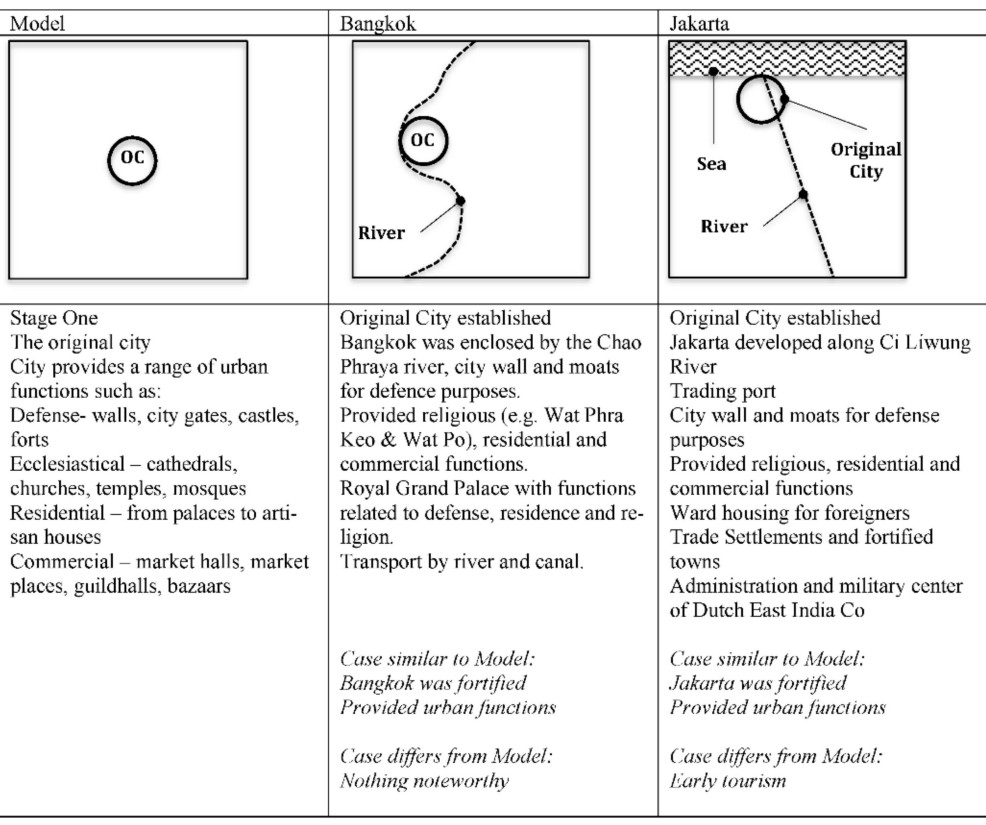

**Figure 1.** Temporal-spatial comparative evolution of model and cases of Bangkok and Jakarta at Stage 1. Source: After Ashworth and Tunbridge [14], after Burton [32] and authors.

*Stage Two*

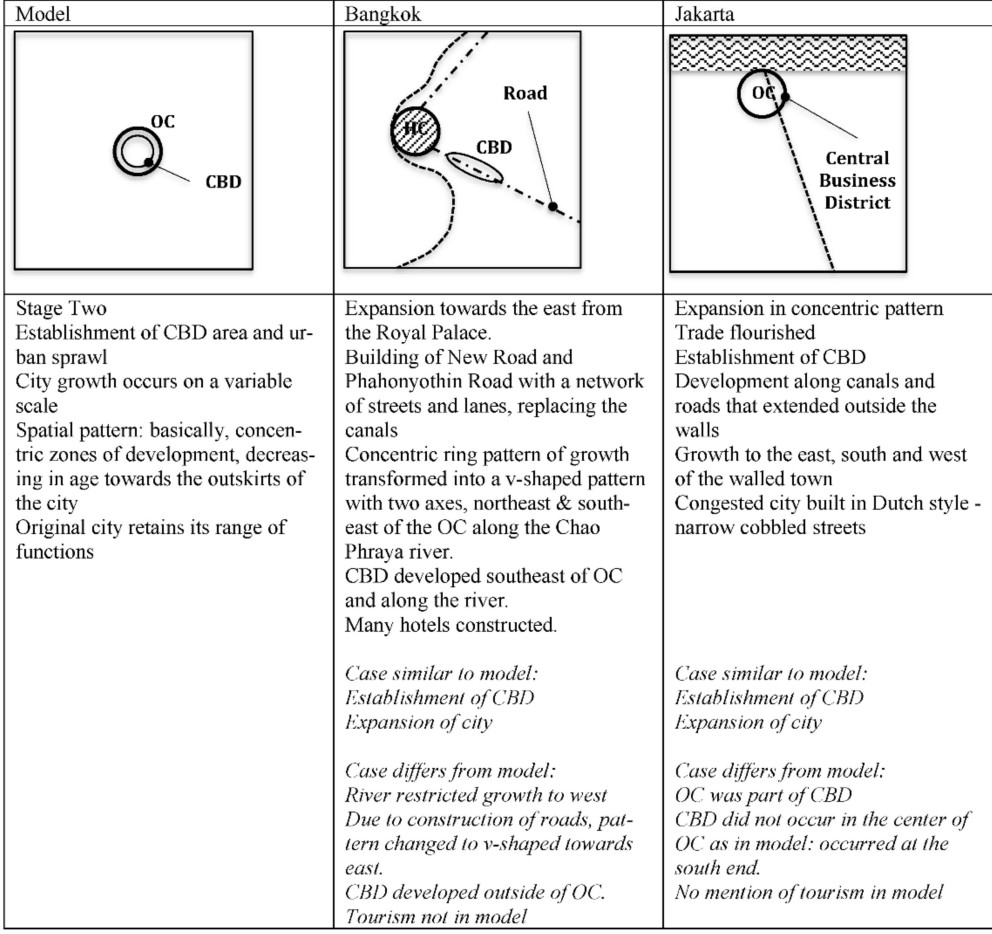

| Model | Bangkok | Jakarta |
|---|---|---|
| Stage Two<br>Establishment of CBD area and urban sprawl<br>City growth occurs on a variable scale<br>Spatial pattern: basically, concentric zones of development, decreasing in age towards the outskirts of the city<br>Original city retains its range of functions | Expansion towards the east from the Royal Palace.<br>Building of New Road and Phahonyothin Road with a network of streets and lanes, replacing the canals<br>Concentric ring pattern of growth transformed into a v-shaped pattern with two axes, northeast & southeast of the OC along the Chao Phraya river.<br>CBD developed southeast of OC and along the river.<br>Many hotels constructed.<br><br>*Case similar to model:*<br>*Establishment of CBD*<br>*Expansion of city*<br><br>*Case differs from model:*<br>*River restricted growth to west*<br>*Due to construction of roads, pattern changed to v-shaped towards east.*<br>*CBD developed outside of OC.*<br>*Tourism not in model* | Expansion in concentric pattern<br>Trade flourished<br>Establishment of CBD<br>Development along canals and roads that extended outside the walls<br>Growth to the east, south and west of the walled town<br>Congested city built in Dutch style - narrow cobbled streets<br><br>*Case similar to model:*<br>*Establishment of CBD*<br>*Expansion of city*<br><br>*Case differs from model:*<br>*OC was part of CBD*<br>*CBD did not occur in the center of OC as in model: occurred at the south end.*<br>*No mention of tourism in model* |

**Figure 2.** Temporal-spatial comparative evolution of model and cases of Bangkok and Jakarta at Stage 2. Source: After Ashworth and Tunbridge [14], after Burton [32] and authors.

Originating at the mouth of the Ci Liwung river, Jakarta became an important trading and commercial center through Indonesia's colonial era and increasingly so as the capital of an independent sovereign nation. Coastal proximity and a river port were major determinants for urban growth. Bangkok also had its origin as a port on the Chao Phraya river, though it was situated some distance inland from the sea.

The specific physical geography of these cases heavily influenced their urban forms over time. The growth of Jakarta was restricted by the sea to its north. Consequently, the city expanded to the south along both banks of the Ci Liwung where the modest scale of this river posed few limitations for development. The urban development of Bangkok was also restricted but by the Chao Phraya river, which was a very large waterway, much larger than the Ci Liwung river. Here the city mostly expanded to the east of the Chao Phraya with considerably less urbanization on its west bank.

Both cases had fortified, walled origins, which mirrors the original city (OC) in stage one of the modified Ashworth and Tunbridge model. In stage two, each of the cases had urbanized and possessed a central business district (CBD). The urban develop patterns were clearly influenced by the presence of a major river for Bangkok and the sea for Jakarta; unlike the neat concentric geometric pattern represented in the model. The CBDs of both cases developed, mainly, outside of the OC, unlike the model that had its CBD entirely within the OC. Interestingly, tourism occurred in both cases at stage two when the model does not mention this function at this stage, which seems to us to be an aberration in the

model as it is difficult accept that tourism did not exist for this stage in Europe but it did in Asia.

***Stage Three***

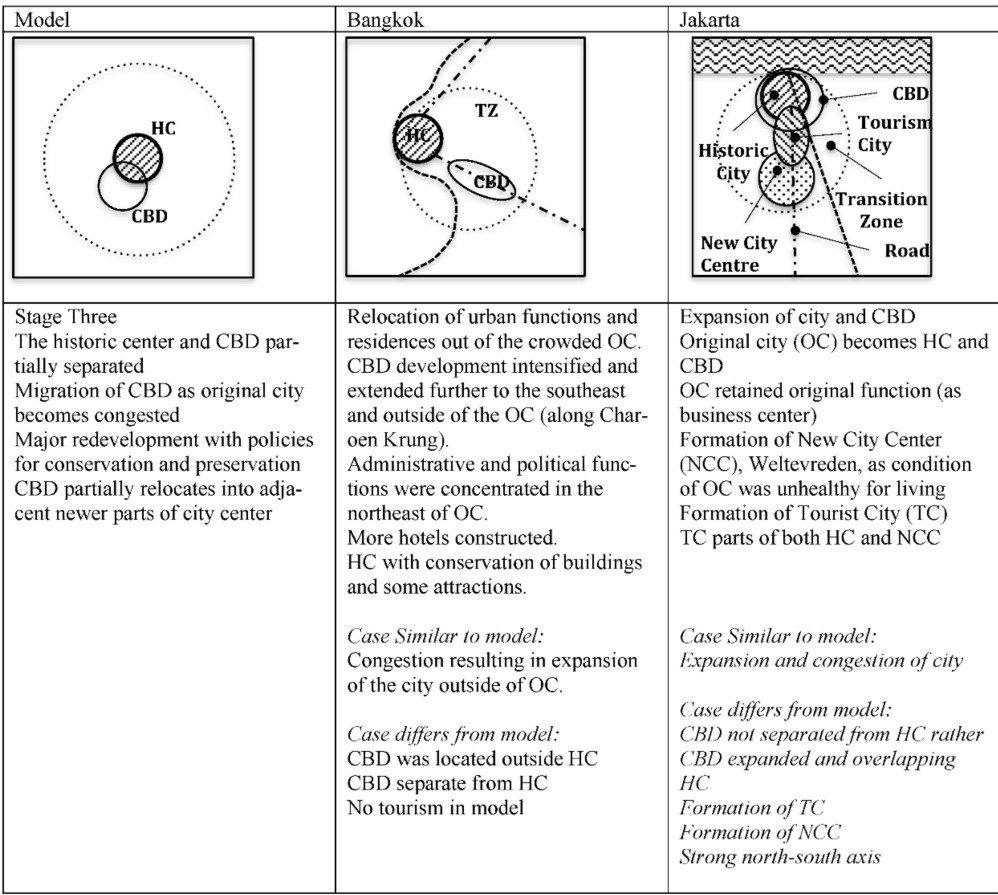

| Model | Bangkok | Jakarta |
|---|---|---|
| Stage Three<br>The historic center and CBD partially separated<br>Migration of CBD as original city becomes congested<br>Major redevelopment with policies for conservation and preservation<br>CBD partially relocates into adjacent newer parts of city center | Relocation of urban functions and residences out of the crowded OC.<br>CBD development intensified and extended further to the southeast and outside of the OC (along Charoen Krung).<br>Administrative and political functions were concentrated in the northeast of OC.<br>More hotels constructed.<br>HC with conservation of buildings and some attractions.<br><br>*Case Similar to model:*<br>*Congestion resulting in expansion of the city outside of OC.*<br><br>*Case differs from model:*<br>CBD was located outside HC<br>CBD separate from HC<br>No tourism in model | Expansion of city and CBD<br>Original city (OC) becomes HC and CBD<br>OC retained original function (as business center)<br>Formation of New City Center (NCC), Weltevreden, as condition of OC was unhealthy for living<br>Formation of Tourist City (TC)<br>TC parts of both HC and NCC<br><br>*Case Similar to model:*<br>*Expansion and congestion of city*<br><br>*Case differs from model:*<br>*CBD not separated from HC rather*<br>*CBD expanded and overlapping*<br>*HC*<br>*Formation of TC*<br>*Formation of NCC*<br>*Strong north-south axis* |

**Figure 3.** Temporal-spatial comparative evolution of model and cases of Bangkok and Jakarta at Stage 3. Source: After Ashworth and Tunbridge [14], after Burton [32] and authors.

With expansion of the case cities, urbanization intensified the development of their respective CBDs. The clear disconnection between OC and CBD in both cases—with complete separation for Bangkok—suggests weak links between historic attractions and formal commerce. This is contrary to stage three of the model that has the CBD within the OC. In this context it is worth recalling that these historic attractions were often the repository of government functions. Some of these buildings continue to have formal governmental functions related, in Bangkok, to the reigning king of Thailand.

Early in the life of both cases, urban development extended away from the OC along well-defined axes that were important land transportation links to their hinterlands. For Jakarta, this was to the south. Bangkok had two axes-one to the north and the other to the east that ran parallel to the Chao Phraya river. These major land axes that developed over stages three and four became major generators of urban development, as proximity to transport was an advantage for business. Earlier transport for Bangkok had been by water but by stage four it had become mainly by land.

*Stage Four*

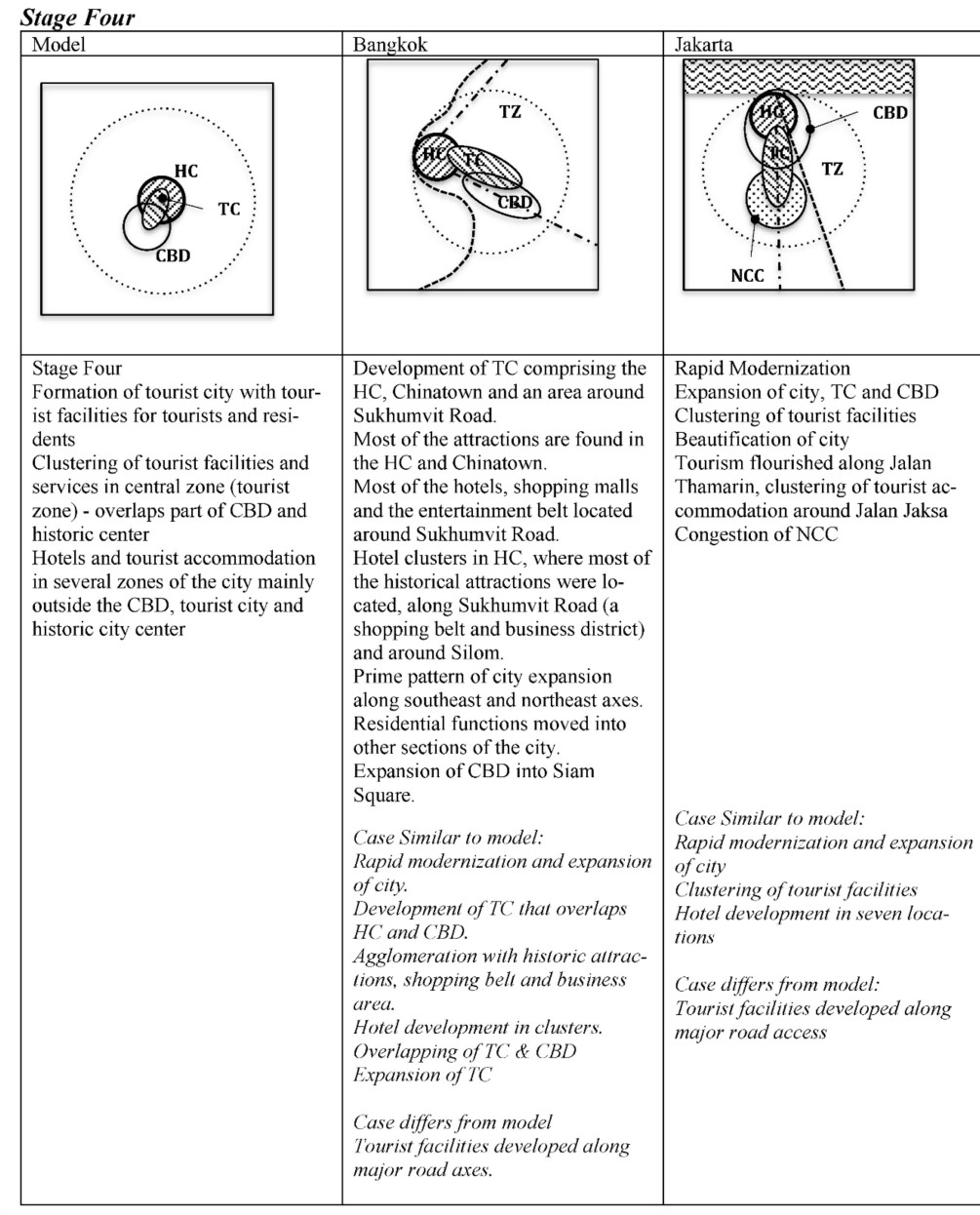

| Model | Bangkok | Jakarta |
|---|---|---|
| Stage Four<br>Formation of tourist city with tourist facilities for tourists and residents<br>Clustering of tourist facilities and services in central zone (tourist zone) - overlaps part of CBD and historic center<br>Hotels and tourist accommodation in several zones of the city mainly outside the CBD, tourist city and historic city center | Development of TC comprising the HC, Chinatown and an area around Sukhumvit Road.<br>Most of the attractions are found in the HC and Chinatown.<br>Most of the hotels, shopping malls and the entertainment belt located around Sukhumvit Road.<br>Hotel clusters in HC, where most of the historical attractions were located, along Sukhumvit Road (a shopping belt and business district) and around Silom.<br>Prime pattern of city expansion along southeast and northeast axes.<br>Residential functions moved into other sections of the city.<br>Expansion of CBD into Siam Square.<br><br>*Case Similar to model:*<br>*Rapid modernization and expansion of city.*<br>*Development of TC that overlaps HC and CBD.*<br>*Agglomeration with historic attractions, shopping belt and business area.*<br>*Hotel development in clusters.*<br>*Overlapping of TC & CBD*<br>*Expansion of TC*<br><br>*Case differs from model*<br>*Tourist facilities developed along major road axes.* | Rapid Modernization<br>Expansion of city, TC and CBD<br>Clustering of tourist facilities<br>Beautification of city<br>Tourism flourished along Jalan Thamarin, clustering of tourist accommodation around Jalan Jaksa<br>Congestion of NCC<br><br><br><br><br><br>*Case Similar to model:*<br>*Rapid modernization and expansion of city*<br>*Clustering of tourist facilities*<br>*Hotel development in seven locations*<br><br>*Case differs from model:*<br>*Tourist facilities developed along major road access* |

**Figure 4.** Temporal-spatial comparative evolution of model and cases of Bangkok and Jakarta at Stage 4. Source: After Ashworth and Tunbridge [14], after Burton [32] and authors.

The tourism city (TC) emerges at stage four of the model and this was also found to occur at this stage for both cases, though the model presents a neat and compact TC not found in either Bangkok or Jakarta. By stage five, both cases had become major metropolises with significant tourism functions, which reflects stage five as presented in the model with the TC expanding into the historic core and modern CBD. The creation of new tourist attractions (both historic and non-historic) occurred in both the cases in response to ever increasing tourism arrival numbers.

*Stage Five*

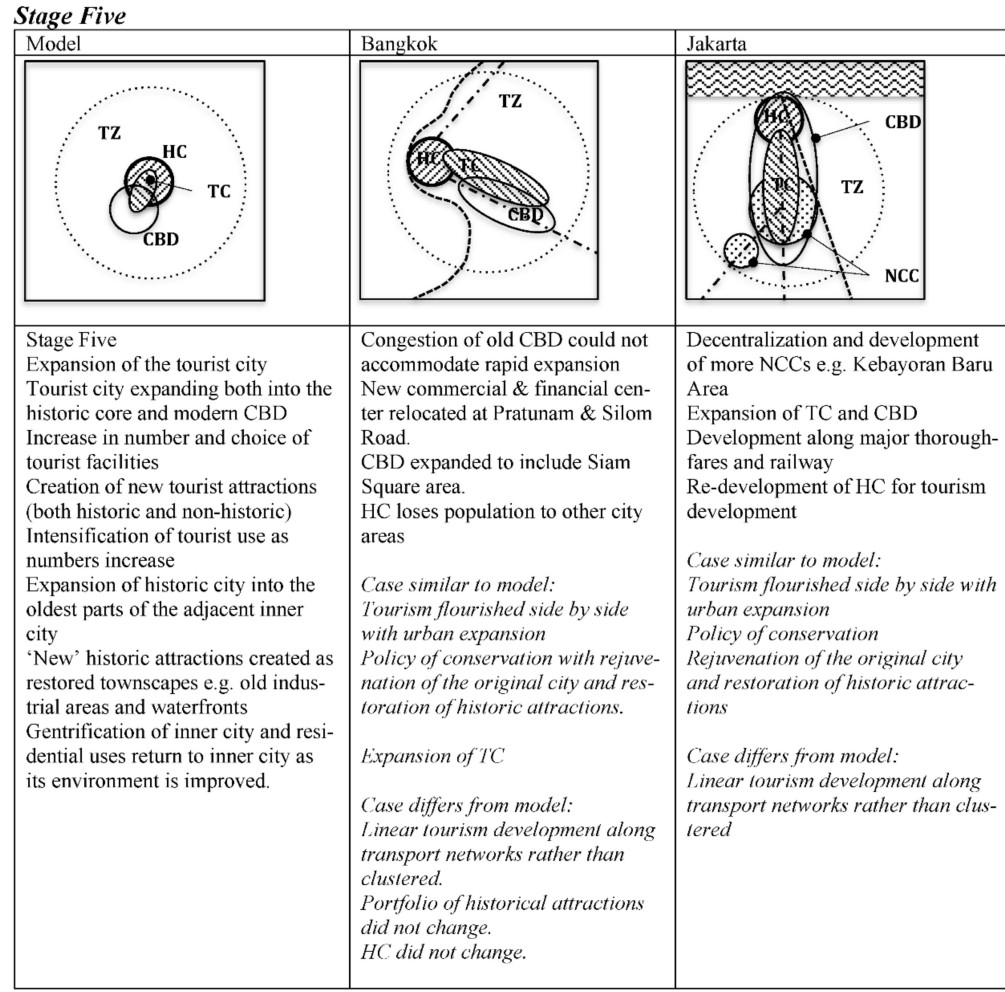

| Model | Bangkok | Jakarta |
|---|---|---|
| Stage Five<br>Expansion of the tourist city<br>Tourist city expanding both into the historic core and modern CBD<br>Increase in number and choice of tourist facilities<br>Creation of new tourist attractions (both historic and non-historic)<br>Intensification of tourist use as numbers increase<br>Expansion of historic city into the oldest parts of the adjacent inner city<br>'New' historic attractions created as restored townscapes e.g. old industrial areas and waterfronts<br>Gentrification of inner city and residential uses return to inner city as its environment is improved. | Congestion of old CBD could not accommodate rapid expansion<br>New commercial & financial center relocated at Pratunam & Silom Road.<br>CBD expanded to include Siam Square area.<br>HC loses population to other city areas<br><br>*Case similar to model:*<br>*Tourism flourished side by side with urban expansion*<br>*Policy of conservation with rejuvenation of the original city and restoration of historic attractions.*<br><br>*Expansion of TC*<br><br>*Case differs from model:*<br>*Linear tourism development along transport networks rather than clustered.*<br>*Portfolio of historical attractions did not change.*<br>*HC did not change.* | Decentralization and development of more NCCs e.g. Kebayoran Baru Area<br>Expansion of TC and CBD<br>Development along major thoroughfares and railway<br>Re-development of HC for tourism development<br><br>*Case similar to model:*<br>*Tourism flourished side by side with urban expansion*<br>*Policy of conservation*<br>*Rejuvenation of the original city and restoration of historic attractions*<br><br>*Case differs from model:*<br>*Linear tourism development along transport networks rather than clustered* |

**Figure 5.** Temporal-spatial comparative evolution of model and cases of Bangkok and Jakarta at Stage 5. Source: After Ashworth and Tunbridge [14], after Burton [32] and authors.

## 5. Conclusions

The cases of Bangkok and Jakarta were evaluated comparatively with the modified Ashworth and Tunbridge model. Both cases reflected and reinforced much of the theory of the evolution of tourism in the historic city. Despite the varying geographical, historic, social, political, and other contexts, we found considerable commonality of evolution with this theory. Despite the uniqueness of origin, society, and economic imperatives of each case, many common patterns of urban development were identified. In addition, however, some variations were also identified which may not be explained solely by restriction or opportunity of local geography. Nevertheless, the resulting urban paradigm is clear, though the conclusions are limited by the fact that only two cases were analyzed.

To summarize, the major findings of our research support the notion that the general spatial transitions over time, as recorded in the Ashworth and Tunbridge model, were valid for these two cases, though it has to be borne in mind that the model is a generalization. Consequently, the physical geographic features of the cases, such as river and coast, significantly influenced the forms of their individual evolutions. Separately, there were some deviations from the model found in Bangkok and Jakarta. Tourism was clearly a component of the economic activities of both cases at an early stage, something that was not reflected in the model. Moreover, the CBD areas of both Bangkok and Jakarta developed mainly externally from the original city and not integrally with the original city. Over time, a definite disconnection of original city and CBD was evident in both Bangkok and Jakarta, which suggests a weak link between historic attractions and the

newer non-tourism commence. Urban development of both cases expanded along primary land axes leading from the original cities. These axes became major generators during their latter stages of urban development. As with the model, the tourism city overlapped the original cities and their respective CBD areas but given the separation of these two areas in Bangkok and Jakarta, the tourism cities became extensive in area.

It is accepted also that both cases are now megacities so our conclusions may not be applicable for smaller cities. Another aspect for further research is the advent of Airbnb and similar forms of tourist accommodation which will possibly be a force for future change in all cities that have a significant tourism economy. Del Romero Renau [48] highlights these dynamics for the historic city of Valencia, Spain, where change in the accommodation structure is underway as propelled by Airbnb.

The research provides some clarification on the temporal urbanization of historic cities in Asia. That tourism has been and is likely to continue to be a prominent feature of historic cities is widely accepted, thus the insights from the present research should provide municipal policy makers, urban planners and private sector developers and operators a better understanding of the spatial dynamics of tourism in urban settings.

It follows that there is the potential for further comparisons with other cases such as Ho Chi Min City, Hong Kong, and Singapore, thus strengthening case comparison generalizations. Future research may combine our analyses to prepare a common model for the Asian historic city. Such a model would hopefully strengthen the general understanding of the evolution of tourism as an economic, social, environmental and institutional entity that reinforces common temporal planning within urban developments.

The challenges faced by urban planners and developers are manifold, where the ultimate goal is arguably sustainable, livable cities [7,8,49]. Our research is important as it contributes to a better understanding of tourism within the complex dynamics of temporal city development. It does this for historic cities in Asia where tourism is a significant economic driver. Urbanization is expanding in Asia so the importance of urban tourism in this region is likely to increase where the social, economic and spatial implications are significant [6,50]. Understanding the evolution of urban tourism development helps to equip tourism stakeholders in meeting the demand and supply aspects of tourism development for a more sustainable tourism development.

In conclusion, the challenges of historic urban tourism in Southeast Asia vary from those of other geographical regions. The new technologies related to greater verticality, mass transit systems and greening present opportunities but raise questions on their sustainable impact on tourism in cities. This research suggests that urban tourism needs to be holistically and constantly evaluated within a multi-dimensional framework, which is critical with ever-increasing scale of tourism urbanization and integration with globalization.

**Author Contributions:** Conceptualization, J.L.T.O. and R.A.S.; methodology, J.L.T.O. and R.A.S; formal analysis, J.L.T.O. and R.A.S.; investigation, J.L.T.O. and R.A.S.; writing—original draft preparation, J.L.T.O. and R.A.S.; writing—review and editing, J.L.T.O. and R.A.S.; visualization, J.L.T.O. and R.A.S. All authors have read and agreed to the published version of the manuscript.

**Funding:** This research received no external funding.

**Acknowledgments:** The authors gratefully acknowledge Chiang Chew Hong Cindy, Kang Hui Ling and Yew Hew Mei, students at the Nanyang Technological University, for their contribution to the research findings for the case of Bangkok, Thailand.

**Conflicts of Interest:** The authors declare no conflict of interest.

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
