# Peer review of "Modelling Urban Tourism in Historic Southeast Asian Cities"

_urbansci, doi:10.3390/urbansci5020038_

Round 1

Reviewer 1 Report

SUggestions:

  • cut out the descriptions of traditional models (Miossec, Butler....) and start the second section (modelling urban tourism) from "One urban tourism model has been developed by Smith (1991)....";
  • write the major findings in the conclusion section. It would be useful to summarize the most interesting findings, even with the aim of testing them in other cities or urban regions

Detailed suggestions please see attachment.

Author Response

Thank you for your insightful and helpful comments, which have been accepted. 

Please see the two (2) attached files.

Reviewer 2 Report

In recent years, studies on urban tourism have registered a significant increase. In the international literature there are numerous studies and cases of considerable importance that must be mentioned. The method of analysis is good. Introduction The introduction should further highlight the importance of this study. Authors should justify the importance of this study based on a strong review of the literature and other sources. A literature review paragraph is recommended to fill this gap.
It is also advisable to replace the wikipedia source with authoritative Thai and Indonesian sources
Findings Authors should discuss the findings and how they can be interpreted from the perspective of previous studies and working hypotheses. The findings and their implications should be discussed in the broadest possible context. Future research directions may also be mentioned. Conclusion Emphasize the importance of this study based on the significant results.  

Author Response

(The authors gave the same response as above.)

Reviewer 3 Report

This is a very interesting piece of research and has a contribution to make to urban tourism research. More references from recent city tourism works are needed - such as found in the Routledge Handbook of Tourism Cities. 

The limitations of the research need to be stated in more detail, as do the conclusions. What are the practical implications of the findings for Bangkok and Jakarta, and other Asian cities? Also the contribitions to knowledge could also be expanded.

Bangkok is rated as the leading city destination in the world by Mastercard; yet this and other city rating schemes are not mentioned. It was also surprising that the traffic congestion in Jakarta was not discussed.

Author Response

(The authors gave the same response as above.)

Round 2

Reviewer 3 Report

The revisons were very appropriate and are well constructed. This is a useful contribution to the literature on urban tourism.